# Factors Associated with Limited Vaccine Literacy: Lessons Learnt from COVID-19

**DOI:** 10.3390/vaccines10060865

**Published:** 2022-05-28

**Authors:** Michelle C. Engelbrecht, N. Gladys Kigozi, J. Christo Heunis

**Affiliations:** Centre for Health Systems Research & Development, Faculty of the Humanities, University of the Free State, P.O. Box 339, Bloemfontein 9300, South Africa; kigozign@ufs.ac.za (N.G.K.); heunisj@ufs.ac.za (J.C.H.)

**Keywords:** COVID-19, vaccine literacy, South Africa

## Abstract

Compared to many other developed countries, South Africa has a lower uptake of COVID-19 vaccinations. Although not widely researched, there is evidence that vaccine literacy (VL) is positively associated with vaccination uptake. Therefore, this study aimed to assess levels of VL among the adult population in South Africa, as well as to identify factors associated with limited VL. A cross-sectional, anonymous online survey was conducted during September 2021. The survey, which included the standardized Health Literacy about Vaccination in adulthood (HLVa) Scale, was widely advertised, yielding a total of 10,466 respondents. The average scores for the two HLVa sub-scales were relatively high: functional (M = 2.841, SD 0.799) and interactive-critical (M = 3.331, SD 0.559) VL. A proposed ‘limited’ VL score (score value ≤ 2.50) was observed in 40% of respondents for functional literacy and 8.2% of respondents for interactive-critical literacy. The main factors associated with limited VL included lower levels of education, lower socio-economic status, not being vaccinated against COVID-19, self-identifying as Black/African or Colored (i.e., people of mixed ethnic descent), having poorer health, and being a woman. The significant association between VL and vaccination uptake provides an impetus for policy makers such as the South African Department of Health to promote VL in the attempt to increase COVID-19 vaccination uptake.

## 1. Introduction

Vaccinations are one of the greatest contributions to global health and development, saving millions of lives each year [1,2]. The development of vaccines has seen smallpox and rinderpest, two major infectious diseases, eradicated. Polio has also almost been eradicated and there is significant success in controlling outbreaks of measles [1]. Despite proven success and scientific evidence, there are still people who refuse to vaccinate, and vaccine hesitancy (being unsure about getting a vaccine) has become a widespread issue. Over the past two decades, vaccine hesitancy has been linked to vaccine preventable disease outbreaks such as measles in the United States of America (USA) and Europe [3,4]. In 2019, the World Health Organization (WHO) listed vaccine hesitancy as one of the top 10 threats to global health [5].

As of 21 April 2022, there were 505,035,185 confirmed COVID-19 cases globally, 8,691,975 cases in Africa and 3,746,424 cases in South Africa. In terms of mortality, 6,210,719 deaths were reported worldwide, 171,457 in Africa and 99,681 in South Africa [6]. During the first year of the pandemic people relied on non-pharmaceutical interventions such as mask wearing, social distancing, and sanitizing to prevent the spread of the virus. With the subsequent development of several successful vaccines the shift is to vaccinate as many people as possible to achieve herd immunity (indirect protection from an infectious disease which occurs when a population is immune either due to vaccination or immunity developed through earlier infection), while still applying non-pharmaceutical measures to prevent the spread of the virus [7]. Despite the availability of COVID-19 vaccines, people remain hesitant to vaccinate. As of 21 April 2022, worldwide, 58.75 persons per 100,000 population were fully vaccinated, with a much lower figure of 31.37 people per 100,000 fully vaccinated in South Africa [6]. On 6 April 2022, the Africa Centers for Disease Control and Prevention reported that only 15.85% of African countries’ populations were fully vaccinated [8]. Reasons why people choose not to vaccinate are complex and do not center solely on vaccine-related concerns, but also include personal and societal level influences [3,4]. One of the main reasons for not accepting COVID-19 vaccines is reportedly the uncertainty about the safety and efficacy of these vaccines [9,10,11,12].

A national USA survey found the people who thought COVID-19 vaccines were unsafe, were less willing to be vaccinated, knew less about the virus, and were more likely to believe COVID-19 vaccine myths [11]. Misinformation about COVID-19 is prolific [13,14], as for the first time in history, technology and social media are being used on a massive scale to keep the public informed about this pandemic. WHO [15] refers to this abundance of information as an “infodemic”, which includes the deliberate spread of mis- and disinformation to undermine the public health response to the pandemic. Research has found that the ability to detect fake information about COVID-19, is associated with the intention to have the vaccine [16]. This emphasizes the importance of health literacy, particularly within the context of COVID-19 given the vast amount of contradictory information circulating particularly on social media. Health literacy enables people to understand the reasons behind medical recommendations and take cognizance of the possible outcomes of their actions [17].

Vaccine literacy (VL) emanates from the concept of health literacy, which is defined as the “degree to which people have the capacity to obtain, process, and understand basic health information and services to make appropriate health decisions” [18] (p. 228). According to the same author, vaccine literacy does not just include knowledge about vaccines, but is also about “developing a system with decreased complexity to communicate and offer vaccines as sine qua non of a functioning health system”. International research has shown that VL has increased the intent to have COVID-19 vaccines [18,19,20], however, barriers to VL exist and include: low educational levels, lack of information access, lack of digital literacy, lack of valid information, and cultural perspectives [21]. Given the lower COVID-19 vaccination uptake in South Africa (vs. other developed countries) and considering the positive association between VL and vaccination uptake, this study assessed COVID-19 functional (i.e., the ability to read and understand information) and interactive-critical (i.e., the ability to engage with information and use it to make decisions) VL among the adult population in South Africa and identified factors associated with limited VL. In order to ensure that the VL Scale measured VL, the construct validity and internal consistency of the Scale were tested.

## 2. Materials and Methods

### 2.1. Design, Setting, and Sample

A cross-sectional, anonymous online survey was conducted among the adult population (i.e., persons 18 years and older) in South Africa during September 2021. At this point, the country was experiencing the tail-end of the third wave of COVID-19 infections, mainly driven by the Delta variant. According to the National Institute for Communicable Diseases [22] (p. 3), a wave is defined as “the period from when the COVID-19 incidence is equal to or greater than 30 cases per 100,000 persons until the weekly incidence equal to or below 30 cases per 100,000 persons”. At this point, approximately 8.6 million people (more than one-fifth of the adult population) were fully vaccinated in the country [23].

The survey was advertised on the University of the Free State’s social media platforms (i.e., Facebook and Twitter) and the data-free Moya application [24]. Moya is a mobile messaging app that allows users to communicate without incurring data costs. The app, developed in South Africa by biNu and released in 2018, provides data-free text messaging that works even when a mobile user has no airtime or data balance on their smartphone device. Moya has over five million monthly active users and sends messages to all its subscribers, and by clicking on the link, interested persons were taken to the survey. A total of 10,466 adults participated in the survey, with slightly less than half accessing the survey via the Moya app (n = 5000), while the remaining respondents were exposed to social media advertisements or word of mouth (n = 5466).

### 2.2. Research Instrument and Data Collection

The questionnaire was available in seven of the most frequently spoken languages in South Africa—Zulu, Xhosa, Afrikaans, English, Tsepedi, Setswana and SeSotho—and comprised the following sections: socio-demographic information, including gender, age, race, education, location, religious affiliation, marital status, employment status and income level; health and COVID-19; VL; and uptake of COVID-19 vaccines.

VL was measured using the Health Literacy about Vaccination in adulthood (HLVa) Scale [25]. The Scale comprises 12 items and two scales measuring functional and interactive-critical VL. Functional VL refers to language capabilities such as basic reading skills and comprehension of read content. Interactive-critical VL focuses on more advanced cognitive skills such as problem solving and decision making. Four items assessed functional VL and nine items measured interactive-critical VL. Four-point Likert scales were used to rate the responses for the items measuring functional VL (4—never; 3—rarely; 2—sometimes; 1—often) and interactive-critical VL (1—never; 2—rarely; 3—sometimes; 4—often). This Scale has been used in previous studies to determine COVID-19 VL [12,17,19,20]. The Scale has been found to have suitable psychometric characteristics for the subjective measure of VL in individuals and populations studies [17,20].

The online questionnaire was hosted on a data-free website, to avoid respondents using their own data to participate in the study, and was open for one month (September 2021). During this period, there was continuous marketing of the survey to ensure that as many people as possible were aware of the study. Before accessing the information leaflet and informed consent document, the potential respondents were requested to select the language that they wished to complete the questionnaire in and to confirm that it was the first time that they were participating in the survey.

### 2.3. Data Analysis

Data was analyzed in IBM SPSS version 27 (IBM Corp., New York, NY, USA) [26] and Python [27]. The data was described using frequency counts and percentages for categorical variables and means and standard deviations for continuous variables. Confirmatory factor analysis was undertaken to determine the construct validity of the VL Scale. Cronbach’s alpha was used to test the internal consistency of the VL sub-scales. A *t*-test was used to determine if there was a significant difference in scores on the two-VL sub-scales. Binomial logistic regression was used to determine which factors were significantly associated with limited functional and interactive-critical VL. Although no cut-off value has been established for the VL Scale, Biasio et al. [17] proposed a ‘limited’ VL score when the value of the sub-scale scores is ≤2.50. All assumptions for binomial logistic regression were met. Independent variables included in the model were: age; gender (male and female, LGBTQI was dropped as there were only 11 respondents in this category); race (Black/African, White, Colored (i.e., multiple ethnic groups with mixed ancestors from Africa, Asia, and Europe); education (no formal, primary school, secondary school, tertiary education); monthly income (ZAR 24,000+/USD 1600+, ZAR 12,001–24,000/USD 801–1600, ZAR 6001–12,000/USD 401–800, ZAR 3001–6000/USD 201–400, ZAR 3000 and less/USD 200 and less); health status (good, fair, poor), COVID-19 vaccination status (yes, no).

### 2.4. Ethics

Ethical clearance was obtained from the Health Sciences Research Ethics Committee (HSREC) at the University of the Free State (UFS-HSD2021/0750/3108). Participation in the study was voluntary. Before accessing the questionnaire, the potential respondents were provided with sufficient information regarding the study in order to make an informed choice of whether or not to participate. This included information pertaining to the aim of the study, the responsible researchers, nature of the questions, and estimated time required to complete the questions. It was also indicated that while respondents would receive no remuneration for their participation, for every 1000 completed questionnaires, 100 masks would be donated to a previously disadvantaged primary school, to a maximum of 1000 masks. Information was also available for respondents requiring vaccine-related information, as well as the contact details for free counselling services if they experienced any form of emotional distress due to COVID-19. The interested respondents were then required to indicate that they were 18 years and older, and that they consented to participate in the study. Only after that was completed, were they redirected to the questionnaire. The data was secured in encrypted files.

## 3. Results

### 3.1. Demographic and Background Information

Approximately two-thirds of the respondents were male (65.1%), which is different from the South African demographic profile where there is almost a 50/50 split of males and females in the country [28]. The average age was 43.5 years (SD 13.68). Most respondents self-identified as Black/African (63.4%) followed by White (23.8%). Approximately three-quarters were Christian (74.6%). The majority (60.4%) were unemployed. Almost a quarter of the employed earned less than ZAR 3000 (USD 207) per month (24.5%). Provinces with the most respondents—Gauteng (30.1%), KwaZulu-Natal (15.8%), and the Western Cape (14.4%)—are also the most populous provinces in South Africa [28] (see Table 1 for demographic and background variables).

Approximately 15% of respondents indicated that they had had COVID-19 (14.9%), while 9.3% were uncertain. Of the respondents who had not had COVID-19, 58% did not think that they were likely to be infected in the future, while 32.3% were not sure. About three in five respondents were not vaccinated against COVID-19 (60%) and of these respondents, 29.2% were uncertain they would be vaccinated in future, while 15.5% indicated that they would not have a COVID-19 vaccine.

### 3.2. Vaccine Literacy

The primary sources of information about COVID-19 vaccines included television (21.6%), social media (20.6%), radio (17.4%), family and friends (14.0%), newspapers (13.7%) and, lastly, healthcare workers (12.2%). Just more than two-thirds of respondents (68.2%) wanted more information about COVID-19 vaccines.

Confirmatory factor analysis was used to establish the construct validity of the HLVa Scale [25] for South African adults (i.e., 18 years and older). Factor analyses in previous samples have shown a two-factor model for the Scale (i.e., functional literacy and interactive-critical literacy). All items on the Scale were scored on a 4-point Likert scale and were treated as ordinal variables in the analysis. Exploratory data analysis revealed substantial deviations from normality for some of the variables, which is not surprising given the ordinal nature of the data. Descriptive statistics for all observed variables are provided in Table 2.

The model was fitted using Diagonally Weighted Least Squares estimation due to the ordinal nature of the data. The model fit was good, with a Tucker–Lewis Index (TLI) of 0.900, a Comparative Fit Index (CFI) of 0.918, and a Root Mean Square Error of Approximation (RMSEA) of 0.077. As expected, the indicators all showed significant positive factor loadings, with standardized coefficients ranging from 0.443 to 0.816 (see Table 3) and were similar to those reported by Biasio et al. [17].

These results are consistent with previous studies [17,20] showing a two-factor latent structure for the HLVa Scale, with the two latent factors underlying the items being Functional VL and Interactive-Critical VL. This finding provides evidence for the construct validity of the HLVa Scale. Furthermore, Cronbach’s alphas of 0.848 and 0.816 were, respectively, reported for the Functional VL and Interactive-Critical VL sub-scales, indicating a sufficient measure of internal consistency [29]. There was a significant difference between the mean scores for functional VL (M = 2.841, SD 0.799) and interactive-critical VL (M = 3.331, SD 0.559), t(10,465) = 51.403, *p* = 0.000.

Although no cut-off value has been established for VL Scale, Biasio et al. [17] proposed a ‘limited’ VL score (score value ≤ 2.50), which was observed in 40% of respondents for functional VL and 8.2% for interactive-critical VL. Binomial regression was run to determine the association between age, gender, race, education, monthly income, overall health status, COVID-19 vaccination status and limited functional VL (Table 4) on the one hand, and interactive-critical VL (Table 5), on the other.

All variables included in the regression model (i.e., age, gender, race, education, monthly income, health and vaccination status) were independently statistically significantly associated with limited functional VL. The multivariate logistic regression model for limited functional VL (Table 4) was statistically significant, implying that the predictors as a set reliably distinguished between persons who had limited functional VL and those who had higher levels of functional VL, X^2^(14) = 398.777, *p* < 0.005. The model explained 14.2% (Nagelkerke R^2^) of the variance in the tendency to have limited functional literacy and correctly classified 68.3% of the cases. After controlling for other variables in the model, five predictor variables were found to be statistically significant (*p* < 0.05): race, level of education, monthly income, overall health status, and COVID-19 vaccination status. With regard to race, compared to persons who self-classified as White, persons who self-identified as Black/African (AOR = 2.804, *p* = 0.000), and Colored (AOR = 2.110, *p* = 0.00) were, respectively, 2.8 and 2.1 times more likely to have limited functional literacy. Compared to persons with a tertiary education, persons with a secondary (AOR = 1.339, *p* = 0.001) or primary/no formal (AOR = 2.795, *p* = 0.00) education were, respectively, 1.3 and 2.8 times more likely to have limited functional literacy. Income was also significantly associated with limited functional VL, with persons earning ZAR 12,001–24,000/USD 801–1600 (AOR = 1.451, *p* = 0.003) being 1.4 times more likely to have a limited functional literacy compared to persons earning more than ZAR 24,000/USD 1600. Persons who classified their overall health as being fair (AOR = 1.484, *p* = 0.000) or poor (AOR = 1.644, *p* = 0.043) were 1.5 and 1.6 times more likely to have limited functional literacy compared to persons who classified their health as being good. Finally, compared to persons who were vaccinated against COVID-19, those not vaccinated (AOR = 1.312, *p* = 0.001) were 1.3 times more likely to have limited functional VL.

Age, gender, race (i.e., Black/African and Colored), education, monthly income (i.e., groups earning more than ZAR 3000/USD 200), poor health, and vaccination status were independently significantly associate with limited interactive-critical VL. The multivariate logistic regression model for interactive-critical VL was statistically significant, implying that the predictors as a set reliably distinguished between persons who had limited interactive-critical VL and those who had higher levels of interactive-critical VL, X2(14) = 94.853, *p* < 0.005. The model explained 6.7% (Nagelkerke R2) of the variance in the tendency to have limited interactive-critical VL and correctly classified 93.6% of the cases. After controlling for other variables in the model, four predictor variables were found to be statistically significant (*p* < 0.05): gender, education, monthly income, and vaccination status. Women were 1.5 times more likely to have limited interactive-critical literacy than men (AOR = 1.461, *p* = 0.009). Compared to persons with a tertiary education, persons with secondary (AOR = 2.078, *p* = 0.000) or primary/no formal (AOR = 2.740, *p* = 0.014) education were 2.1 and 2.7 times more likely to have limited interactive-critical VL. Lower income groups, i.e., those persons earning ZAR 6001–12,000 per month (AOR = 2.068, *p* = 0.012) and ZAR 3001–6000/USD 201–400 per month (AOR = 1.899, *p* = 0.033) were 2.1 and 1.9 times more likely to have limited interactive-critical VL than those earning more than ZAR 24,000/USD 1600 per month. Unvaccinated persons (AOR = 1.543, *p* = 0.005) were 1.5 times more likely to have limited functional interactive literacy than vaccinated persons.

## 4. Discussion

This study assessed COVID-19 functional (the ability to read and understand information) and interactive-critical (the ability to engage with information and use it to make decisions) VL among the adult population in South Africa and identified factors associated with limited VL. The average scores for functional and interactive-critical VL were relatively high and are comparable to those found by Biasio et al. [17] in an online survey among the adult population in Italy. A similar study among Croatian adults reported lower levels of both functional and interactive-critical VL, with a higher mean score for interactive-critical VL [20]. Both these studies took place before the rollout of COVID-19 vaccines, while our research was conducted when South Africa had already vaccinated slightly more than a fifth of the adult population 35 years and older and had just started vaccinating persons 18 years and older [23].

Given the timing of our study, one might have expected even higher scores for functional VL as more information was available about COVID-19 vaccines, as well evidence that the vaccines were protecting individuals from infection. On the negative side, levels of functional VL may have remained lower as information about the vaccines are generally available in English, and respondents whose home language was not English might have struggled to read and understand the information. A higher score for interactive-critical VL suggests that our respondents were engaging with COVID-19 vaccine-related information and using it to make decisions; however, as their functional VL was lower, they might not always have understood the information that they were engaging with. Furthermore, while there is a lot of COVID-19 information, much of this is contradictory (mis)information about vaccines [13], easily available on social media [30,31,32], but also on traditional media [31]. As noted by Abel and McQueen [33] (p. 1612), “In the exploding market of COVID-19 facts and fiction, individuals need to know how to assess critically the information with which they are overwhelmed”. Contradictory (mis)information can be difficult to assess even for people with high literacy scores [17]. The likelihood that some of our respondents engaged with contradictory (mis)information is supported by the finding that they mostly relied on mediums such as television and social media that are less likely to be objective, with far fewer relying on information from healthcare workers, arguably a much more reliable source. Furthermore, despite widespread availability of COVID-19 vaccine information, more than two-thirds of the respondents wanted more information.

A closer examination of the VL scores revealed that while the average scores for functional and interactive-critical literacy were high, the suggested ‘limited’ VL score (cut-off of ≤2.50), categorized 40% of respondents as having limited functional VL compared to 8% of respondents with limited interactive-critical literacy, which is similar to that reported by Baiso et al. [17]. Limited VL is associated with reduced adoption of protective behaviors such as immunization [34]. Factors associated with both limited functional VL and limited interactive VL in our study included: lower levels of education, lower income groups, and not being vaccinated against COVID-19. The association between lower levels of education and VL has been reported in similar studies [20,35] among Croatian adults and autoimmune disease patients, respectively. Correa-Rodríguez et al. [35] also found a relationship between lower income groups and lower levels of interactive-critical VL. The association between socio-economic status, education level, and VL is somewhat expected, as persons with higher levels of education are more likely to have a higher socio-economic status and could be expected to have better access to knowledge and therefore be able to make sense of the information they have. However, Biasio [36] cautions that higher levels of education do not always correspond with the ability to critically interpret information, as information overload can result in persons with higher levels of functional and interactive-critical VL incorrectly assessing and evaluating information.

We found that persons who had not had a COVID-19 vaccine were more likely to have limited functional and interactive-critical VL. Similarly, research indicated that higher levels of functional and interactive-critical VL are associated with the intention to have the COVID-19 vaccine [17,19,20]. Furthermore, Yadete et al. [37] found in their assessment of the acceptability of COVID-19 vaccine booster dose among adult Americans, that that the booster-hesitant group had lower scores for functional, communicative, and critical vaccine literacies. Contrarily, Nath et al. [38] found that VL did not have any influence on young Bangladesh people’s intention to have a COVID-19 vaccination. Despite contradictory evidence as to the role of VL in influencing vaccination uptake, this remains an important area that requires further investigation, especially given the limited research that has been undertaken in this field.

Race and health status are two additional factors associated with limited functional VL. The association between racial groups and limited functional VL may relate to the fact that the illiteracy rate in South Africa is highest among Blacks/Africans and Coloreds [39], which is perpetuated by racial inequalities in the schooling system [40]. Gusar et al. [20] reported lower levels of functional and critical VL among persons who took medication on a daily basis, which supports our finding that persons who considered their health to be “fair” were also more likely to have limited functional VL than persons who considered themselves to be in “good” health. This is interesting, as one would have expected persons who did not consider themselves to be in “good” health to be more inclined to gain information about their health and improve their health and by extension their VL. The final factor that we found to influence VL was gender. Women were more likely to have a limited interactive-critical VL score than men. This finding is contrary to what has been reported in similar research [35], where women were found to have higher interactive-critical VL than men, while other research has found no significant association between gender and levels of interactive-critical VL [20]. This warrants further research to establish why gender differences occur and how they can be addressed.

As of 9 March 2022, 43.16% of adults 18 years and older have been fully vaccinated in South Africa [41], while the goal was to have 70% of the population vaccinated by the end of 2021 [23]. Although COVID-19 cases have significantly dropped in the country, concerns have been raised about an eminent 5th wave of infections. Given the uncertainties surrounding which variant will be the key driver of the 5th wave, and how severe this variant may be, experts continue to emphasize the importance of vaccinating [42]. Based on our findings, it is important to consider VL when promoting the uptake of vaccination. In particular, when aiming to increase VL among the South African population, one should focus on persons with lower levels of education and lower socio-economic status, persons who self-identify as Black/African or Colored, those in poorer health, and women.

Given the widespread reliance on television and social media for COVID-19 information, these and other information channels should regularly be scrutinized to ensure the accuracy of information that is being distributed about vaccines. Gisondi et al. [43] (p. 1) warned that “During the COVID-19 pandemic, substantial attention was initially focused on ensuring the distribution of the vaccines themselves but, unfortunately, not the distribution of reliable information nor the mitigation of harmful misinformation and disinformation”. Social media companies are not responsible for the content that users post; however, there is an expectation that they will enforce “Good Samaritan” practices and self-regulate objectionable content. Despite noble intentions, significant errors are made when moderating information posted on social media [44]. In light of these challenges, in March 2020, the South African government, under the Disaster Management Act (Regulation 11.5), passed regulations that criminalized spreading misinformation about the COVID-19 pandemic on any platform, including SMS, WhatsApp, Twitter, Facebook, Instagram, online videos, and other messaging and networking or social media platforms [45]. Nevertheless, a year later the Acting Minister of Health in South Africa, Mmamoloko Kubayi-Ngubane, again raised concerns that the spread of COVID-19 “fake news” made it difficult to stop the spread of the virus in the country [46]. In this regard, the National Department of Health can play a key role in promoting COVID-19 VL, particularly via television and social media platforms through engaging patients on social media; posting COVID-19 vaccine messages online, but also across different platforms such as television and radio; providing health expertise for social media companies as well as other mass media outlets; and empowering patients to search for reliable and accurate health information in order to make informed choices [43].

The value of our study is that it builds on a limited database of research investigating the concept of VL. However, as with all research, ours too has limitations. Firstly, we used an online survey to collect our data, which could potentially have excluded members of the population who do not own a smart phone, tablet, or computer. Secondly, our sample was not randomly selected and as such caution needs to be exercised when interpreting and generalizing the results. For example, there was about 15% over-representation of men in our sample. This may be a result of men being more likely to own cellular or ‘smart’ phones or personal computers as is the case throughout lower- and middle-income countries [47]. In addition, the racial breakdown of our respondents also differs from the South African population profile—with an over-representation of Whites (i.e., 23.8% of our respondents were White compared to 7.8% in the general population) [28]. Thirdly, the cross-sectional nature of the data, which was collected at one specific time during the pandemic, does not allow for interpretation for causality. Fourthly, the data is self-reported, and as such we are reliant on the participants’ honesty. Finally, as this was a self-administered online questionnaire, we were cognizant of the length of the questionnaire and the time it would take to complete. As such, we were limited in the number of questions and excluded potentially relevant issues such as the language and platform preference for information dissemination that could also have contributed to VL.

## 5. Conclusions

While research on VL, particularly COVID-19 VL, is limited, evidence suggests the importance of studying this concept further given the associations found between VL and uptake of vaccination. In particular, persons with lower levels of education and socio-economic status, who self-identify as Black/African or Colored, with poorer health and women tend to be more at risk for limited levels of VL. The importance of providing accurate and reliable vaccine information cannot be over-emphasized. The National Department of Health can potentially play a significant role in ensuring the accuracy of information circulating on mass media and social media platforms. While results appear to be country-specific, countries with similar socio-economic profiles can draw on this information to identify groups potentially at risk for low VL requiring tailored intervention.

## Figures and Tables

**Table 1 vaccines-10-00865-t001:** Demographic and background variables.

Variable		n	%
Province (n = 10,466)	Gauteng	3153	30.1
	KwaZulu-Natal	1651	15.8
	Western Cape	1510	14.4
	Free State	1174	11.2
	Eastern Cape	919	8.8
	Mpumalanga	637	6.1
	North West	636	6.1
	Limpopo	527	5.0
	Northern Cape	259	2.5
Gender (n = 10,466)	Male	6812	65.1
	Female	3643	34.8
	LGBTIQ	11	0.1
Race (n = 10,466)	Black/African	6636	63.4
	White	2493	23.8
	Colored	1190	11.4
	Asian	147	1.4
Religion (n = 10,466)	Christian	7801	74.6
	Traditional African	1951	18.6
	Muslim	255	2.4
	Other	459	4.4
Education (n = 10,466)	No formal education	54	0.5
	Primary school	130	1.2
	Secondary school	6023	57.5
	Tertiary education	4259	40.7
Employment status (n = 10,466)	Unemployed	6324	60.4
	Employed full-time	2393	22.9
	Employed part-time	999	9.5
	Retired	608	5.8
	Student	142	1.4
Income (n = 3692)	ZAR 3000 and less (USD 200 and less)	904	24.5
	ZAR 3001–6000 (USD 201–400)	712	19.3
	ZAR 6001–12,000 (USD 401–800)	486	13.2
	ZAR 12,001–24,000 (USD 801–1600)	758	20.5
	More than ZAR 24,000 (USD 1600+)	832	22.5

**Table 2 vaccines-10-00865-t002:** Descriptive statistics VL Scale.

Items	Mean	SD	Min	Max
When you read or listen to information about COVID-19 vaccines, are there words you do not know? *	2.69	0.92	1	4
When you read or listen to information about COVID-19 vaccines, are there parts that are difficult to understand? *	2.82	0.93	1	4
When you read or listen to information about COVID-19 vaccines, do you need more time to understand such information? *	2.79	1.01	1	4
When you read or listen to information about COVID-19 vaccines, do you need someone to help you understand such information? *	3.06	0.99	1	4
When you look for information about COVID-19 vaccines, do you consult more than one source of information? **	3.09	0.98	1	4
When you look for information about COVID-19 vaccines, do you find the information you are looking for? **	3.41	0.79	1	4
When you look for information about COVID-19 vaccines, do you have the opportunity to use the information? **	3.22	0.84	1	4
When you look for information about COVID-19 vaccines, do you discuss what you understand about vaccinations with your doctor? **	2.51	1.13	1	4
When you look for information about COVID-19 vaccines, do you discuss what you understand about vaccinations with other people? **	3.32	0.81	1	4
When you look for information about COVID-19 vaccines, do you consider information about your health condition? **	3.37	0.85	1	4
When you look for information about COVID-19 vaccines, do you consider the credibility of the sources? **	3.37	0.82	1	4
When you look for information about COVID-19 vaccines, do you check whether the information is correct? **	3.52	0.77	1	4
When you look for information about COVID-19 vaccines, do you find useful information to make a decision on whether or not to get vaccinated? **	3.34	0.89	1	4

* 1 = often, 4 = never; ** 1 = never, 4 = often.

**Table 3 vaccines-10-00865-t003:** VL Scale factor loadings.

Item	Factor	Estimate	Est. SD	Std. Err	z Value	*p* Value
When you read or listen to information about COVID-19 vaccines, are there words you do not know?	Functional skills	1.000	0.729	-	-	*
When you read or listen to information about COVID-19 vaccines, are there parts that are difficult to understand?	Functional skills	1.124	0.816	0.015	74.738	*
When you read or listen to information about COVID-19 vaccines, do you need more time to understand such information?	Functional skills	1.172	0.783	0.016	72.602	*
When you read or listen to information about COVID-19 vaccines, do you need someone to help you understand such information?	Functional skills	1.069	0.730	0.016	68.317	*
When you look for information about COVID-19 vaccines, do you consult more than one source of information?	Interactive-critical skills	1.000	0.443	-	-	*
When you look for information about COVID-19 vaccines, do you find the information you are looking for?	Interactive-critical skills	1.120	0.611	0.029	38.168	*
When you look for information about COVID-19 vaccines, do you have the opportunity to use the information?	Interactive-critical skills	1.211	0.624	0.031	38.526	*
When you look for information about COVID-19 vaccines, do you discuss what you understand about vaccinations with other people?	Interactive-critical skills	1.162	0.625	0.030	38.553	*
When you look for information about COVID-19 vaccines, do you consider information about your health condition?	Interactive-critical skills	1.140	0.577	0.031	37.179	*
When you look for information about COVID-19 vaccines, do you consider the credibility of the sources?	Interactive-critical skills	1.217	0.644	0.031	38.040	*
When you look for information about COVID-19 vaccines, do you check whether the information is correct?	Interactive-critical skills	1.214	0.682	0.030	39.934	*
When you look for information about COVID-19 vaccines, do you find useful information to make a decision on whether or not to get vaccinated?	Interactive-critical skills	1.260	0.616	0.033	38.298	*

* *p* < 0.05.

**Table 4 vaccines-10-00865-t004:** Factors associated with limited functional VL.

Variables		Unadjusted Odds Ratio (95% CI)	Adjusted Odds Ratio (95% CI)
Age		0.976 (0.973–0.979)	0.994 (0.986–1.001)
Gender	Male (ref)	1	1
	Female	0.902 (0.832–0.979)	0.901 (0.768–1.057)
Race	White (ref)	1	1
	Black/African	3.381 (3.035–3.766)	2.804 (2.264–3.474)
	Colored	2.871 (2.471–3.335)	2.110 (1.631–2.730)
	Asian	1.460 (1.006–2.121)	0.982 (0.516–1.868)
Education	Tertiary (ref)	1	1
	Secondary	1.930 (1.777–2.096)	1.339 (1.127–1.591)
	Primary/no formal	3.654 (2.694–4.955)	2.795 (1.572–4.968)
Monthly income	ZAR 3000 and less (USD 200 and less) (ref)	1	1
	ZAR 3001–6000 (USD 201–400)	1.747 (1.382–2.208)	1.451 (1.137–1.852)
	ZAR 6001–12,000 (USD 401–800)	2.190 (1.695–2.828)	1.315 (0.994–1.741)
	ZAR 12,001–24,000 (USD 801–1600)	3.334 (2.654–4.189)	1.238 (0.927–1.653)
	More than ZAR 24,000 (USD 1600+)	3.781 (3.042–4.699)	1.216 (0.904–1.635)
Health	Good (ref)	1	1
	Fair	1.561 (1.397–1.744)	1.484 (1.210–1.822)
	Poor	2.059 (1.658–2.556)	1.644 (1.015–2.661)
Vaccination status	Vaccinated (ref)	1	1
	Not vaccinated	0.577 (0.532–0.626)	1.312 (1.118–1.539)

**Table 5 vaccines-10-00865-t005:** Factors associated with limited interactive-critical VL.

Variables		Unadjusted Odds Ratio (95% CI)	Adjusted Odds Ratio (95% CI)
Age		0.988 (0.983–0.994)	1.013 (0.999–1.028)
Gender	Male (ref)	1	1
	Female	1.499 (1.301–1.727)	1.461 (1.098–1.945)
Race	White (ref)	1	1
	Black/African	1.644 (1.357–1.991)	1.114 (0.732–1.695)
	Colored	2.057 (1.600–2.644)	1.057 (0.642–1.741)
	Asian	1.113 (0.555–2.232)	1.513 (0.564–4.057)
Education	Tertiary (ref)	1	1
	Secondary	1.985 (1.695–2.326)	2.078 (1.495–2.890)
	Primary/no formal	3.339 (2.197–5.074)	2.740 (1.224–6.135)
Monthly income	ZAR 3000 and less (USD 200 and less) (ref)	1	1
	ZAR 3001–6000 (USD 201–400)	1.484 (0.866–2.543)	1.334 (0.770–2.310)
	ZAR 6001–12,000 (USD 401–800)	2.856 (1.691–4.822)	2.058 (1.169–3.621)
	ZAR 12,001–24,000 (USD 801–1600)	3.497 (2.169–5.638)	1.899 (1.051–3.429)
	More than ZAR 24,000 (USD 1600+)	3.046 (1.904–4.872)	1.667 (0.904–3.074)
Health	Good (ref)	1	1
	Fair	0.943 (0.767–1.159)	0.842 (0.563–1.260)
	Poor	1.469 (1.047–2.061)	1.526 (0.735–3.169)
Vaccination status	Vaccinated (ref)	1	1
	Not vaccinated	1.891 (1.617–2.210)	1.543 (1.141–2.088)

## Data Availability

Data supporting reported results can be requested from the first author.

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
