# Peer review of "Factors Associated with Limited Vaccine Literacy: Lessons Learnt from COVID-19"

_vaccines, 2022, doi:10.3390/vaccines10060865_

Round 1

Reviewer 1 Report

The work done by Michelle etal is well presented however there are some minor comments should be addressed prior acceptance.

1- the author should clarify the research question of the study.

2- in the introduction, please add more recent data specially in different continent Africa

3- In results, about the demographic data, more information could be included.

4- Is there previous disease history.

5- Conclusion need to be rewritten clearly.

Author Response

Dear Reviewer

Factors associated with limited vaccine literacy: lessons learnt from COVID-19 (vaccines-1680414)

Thank you for the valuable comments on our manuscript. Below is an indication of our response to each point raised. In addition, our corrections have been made in track changes in the manuscript.

1 - the author should clarify the research question of the study.

Response: The research question has been reformulated to improve its clarity (lines 77-82):

Given the lower COVID-19 vaccination uptake in South Africa (vs. other developed countries) and considering the positive association between VL and vaccination uptake, this study assessed COVID-19 functional (i.e., the ability to read and understand information) and interactive-critical (i.e., the ability to engage with information and use it to make decisions) VL among the adult population in South Africa and identified factors associated with limited VL.

2 - in the introduction, please add more recent data specially in different continent Africa

Response: Recent (21 April 2022) data specific to Africa has been added:

As of 21 April 2022, there were 505,035,185 confirmed COVID-19 cases globally, 8,691,975 in Africa and 3,746,424 cases in South Africa. In terms of mortality, 6,210,719 deaths were reported worldwide, 171,457 in Africa and 99,681 in South Africa [6] (lines 36-38).

On 6 April 2022, the Africa Centers for Disease Control and Prevention reported that only 15.85% of African countries’ populations were fully vaccinated [8] (lines 48-50).

3 - In results, about the demographic data, more information could be included.

Response: All the demographic data that was collected have been included.

4 - Is there previous disease history.

Response: We did not include a question on previous disease history. We were limited in the number of questions that we could ask as this was an online survey, and respondents are not inclined to complete long questionnaires. This has been acknowledged as a limitation:

Finally, as this was a self-administered online questionnaire, we were cognizant of the length of the questionnaire and the time it would take to complete. As such, we were limited in the number of questions and excluded potentially relevant issues like the language and platform preference for information dissemination that could also have contributed to VL (lines 400-404).

5 - Conclusion need to be rewritten clearly.

Response: The conclusions have been rewritten as follows:

While research on VL, particularly COVID-19 VL, is limited, evidence suggests the importance of studying this concept further given the associations found between VL and uptake of vaccination. In particular, persons with lower levels of education and socio-economic status, who self-identify as Black/African or Coloured, with poorer health and women tend to be more at risk for limited levels of VL. The importance of providing accurate and reliable vaccine information cannot be over-emphasised. The National Department of Health can potentially play a significant role in ensuring the accuracy of information circulating on mass media and social media platforms. While results appear to be country-specific, countries with similar socio-economic profiles can draw on this information to identify groups potentially at risk for low VL requiring tailored intervention (lines 405-416).

Please do not hesitate to contact with me with any other queries.

Yours sincerely

Michelle Englbrecht

Director & Senior Researcher: Centre for Health Systems Research & Development

Reviewer 2 Report

Engelbrecht et al present an analysis of a survey of individuals in South Africa regarding vaccine hesitancy and demographic factors. The manuscript is overall well composed. The discussion and conclusion read a bit long though the sections of the paper are largely appropriately composed. Ethical review was performed though a formal consent process is not described and what is described on page 3 does not sound like a formal process with documentation of consent as implied by the statement on page 11.

My primary concern is that it is difficult to see where the demographics associated with vaccine literacy were first tested for potential as covariates. Therefore the conclusions seem overstated and socially sensitive as they imply race independently is a risk for vaccine literacy issues when this does not seem clear from the analysis. If there are factors influencing directions certain racial communities are taking I think they can be discussed with more sensitivity and discussion of these factors. The section this is discussed on page 10 seems somewhat cursory.

The definition of Coloured, or why it is considered separately from other ethnicities is not clear to an international audience.

Dollar amounts are reported paired with South African Rand but it is not clear which country's dollar this represents.

The discussion and conclusion are more verbose than they need to be and could be simplified.

Author Response

Dear Reviewer

Factors associated with limited vaccine literacy: lessons learnt from COVID-19 (vaccines-1680414)

Thank you for the valuable comments on our manuscript. Below is an indication of our response to each point raised. In addition, our corrections have been made in track changes in the manuscript.

1 - Engelbrecht et al present an analysis of a survey of individuals in South Africa regarding vaccine hesitancy and demographic factors. The manuscript is overall well composed. The discussion and conclusion read a bit long though the sections of the paper are largely appropriately composed. Ethical review was performed though a formal consent process is not described and what is described on page 3 does not sound like a formal process with documentation of consent as implied by the statement on page 11.

Response: Formal consent was obtained from the respondents. Interested persons clicked on the link to the survey. Before they could move on to the questions, there was an information leaflet and consent form. The information leaflet contained the following information: aim of the study, who was undertaking the study, type of questions that would be asked, estimated time it would take to complete the questionnaire, participation is voluntary, who to contact if they needed emotional support, more information about the project and information relating to ethical issues. In addition, we indicated that respondents would not be remunerated for their participation, however we would donate 100 cloth masks to a previously disadvantaged school for every 1,000 competed questionnaires to a maximum of 5,000 masks.  Respondents were required to indicate that they were older than 18 years and consented to participate before they could move on to the questionnaire. This process has been explained in more detail in the manuscript under “2.4 Ethics:”

Before accessing the questionnaire, the potential respondents were provided with sufficient information regarding the study in order to make an informed choice of whether or not to participate. This included information pertaining to the aim of the study, the responsible researchers, nature of the questions and estimated time required to complete the questions. It was also indicated that while respondents would receive no remuneration for their participation, for every 1,000 completed questionnaires, 100 masks would be donated to a previously disadvantaged primary school, to a maximum of 1,000 masks. Information was also available for respondents requiring vaccine-related in-formation, as well as the contact details for free counselling services if they experienced any form of emotional distress due to COVID-19. The interested respondents were then required to indicate that they were 18 years and older, and that they consented to participate in the study. Only after that was completed, were they redirected to the questionnaire. The data was secured in encrypted files (lines 149-161).

2 - My primary concern is that it is difficult to see where the demographics associated with vaccine literacy were first tested for potential as covariates. Therefore the conclusions seem overstated and socially sensitive as they imply race independently is a risk for vaccine literacy issues when this does not seem clear from the analysis. If there are factors influencing directions certain racial communities are taking I think they can be discussed with more sensitivity and discussion of these factors. The section this is discussed on page 10 seems somewhat cursory.

Response: We have indicated the following information to illustrate that the variables included in the regression models were independently significantly associated with limited functional and interactive-critical vaccine literacy. Therefore, they were also included in the multivariate regression models.

All variables included in the regression model (i.e., age, gender, race, education, monthly income, health and vaccination status) were independently statistically significantly associated with limited functional VL (lines 225-227).

Age, gender, race (i.e. Black/African and Coloured), education, monthly income (i.e., groups earning more than R3,000/$200), poor health, and vaccination status) were independently significantly associate with limited interactive-critical VL (lines 252-254).

Race has been discussed with more sensitivity, e.g. ‘lower socio-economic group’ has been replaced with ‘lower socio-economic status’ throughout the manuscript:

The main factors associated with limited VL included lower levels of education, lower socio-economic status, not being vaccinated against COVID-19, self-identifying as Black/African or Coloured, having poorer health and being a woman (lines 17-20).

In particular, when aiming to increase VL among the South African population, one should focus on persons with lower levels of education and lower socio-economic status, persons who self-identify as Black/African or Coloured, those in poorer health and women (lines 360-363).

As this is not a representative sample, we have also mentioned this under the limitations by adding in the following information:

Secondly, our sample was not randomly selected and as such caution needs to be exercised when interpreting and generalising the results. For example, there was about 15% over-representation of men in our sample. This may be a result of men being more likely to own cellular or ‘smart’ phones or personal computers as is the case throughout lower- and middle-income countries [47]. In addition, the racial breakdown of our respondents also differs from the South African population profile – with an over-representation of Whites (i.e., 23.8% of our respondents were White compared to 7.8% in the general population) [28] (lines 395-397).

3 - The definition of Coloured, or why it is considered separately from other ethnicities is not clear to an international audience.

Response: ‘Coloured’ is now defined as “people of mixed ethnic descent” at first use of the term in the in the methods section:

Independent variables included in the model were: age; gender (male and female, LGBTQI was dropped as there were only 11 respondents in this category); race (Black/African, White, Coloured [people of mixed ethnic descent] and Asian); education (no formal, primary school, secondary school, tertiary education); monthly income (R24,000+, R12,001-R24,000, R6,001-R12,000, R3,001-R6,000, R3,000 and less); health status (good, fair, poor), COVID-19 vaccination status (yes, no) (lines 139-145).

4 - Dollar amounts are reported paired with South African Rand but it is not clear which country's dollar this represents.

Response: The dollar amounts have now been indicated as “US$” throughout the manuscript.

5 - The discussion and conclusion are more verbose than they need to be and could be simplified.

Response: The discussion (lines 275-404) and conclusion (lines 405-416)  sections have been cut and simplified as indicated in “track changes.”

Please do not hesitate to contact with me with any other queries.

Yours sincerely

Michelle Englbrecht

Director & Senior Researcher: Centre for Health Systems Research & Development

Reviewer 3 Report

This is a questionnaire study that aimed to identify factors associated with limited vaccine literacy.  The survey collected data from 10,466 participants through an App Moya.  They reached the conclusion that 40% of respondents as having limited functional VL 292 compared to 8% of respondents with limited interactive-critical literacy.  Significant correlations of limited vaccine literacy were identified to be education, socio-economic status, vaccinated or not, race, health, and gender. This study is a well planned and well presented.  However, there are some shortcomings that need to be addressed.  One major concern is the unclear definition between the two criteria: functional and interactive-critical VL. For instance, on lines 271-276: people cannot understand the information appears to more relate to functional not interactive-critical (decision).  It is the author’s discretion to use any criteria in their study, but you need to make it clear the differences between the criteria you used.

Another major concern is the incomplete information to estimate the applicability of the results in this study.  Two pieces of information should be provided as clear as possible: (1) the percentage of the population who have access to the survey APP Moya. (2) has any incentives been used to stimulate the responses; this will give an indication which group are more likely to participate.

Minor concerns:    

  1. The terminology “vaccine uptake” is ambiguous. In drug delivery, this means the efficiency of the drug delivery.  To avoid potential confusion, I suggest changing it to a more straightforward phase, like “vaccination rate” or “vaccination willingness” or sth others.
  2. Lines 54-56: this sentence does not read well. Have you missed “as” before “technology”?
  3. Lines 65-67: it is not clear from this sentence whether the quoted definition is VL or health literacy.
  4. Lines 86-87: this sentence is disconnected from the sentences above. It is suggested to change to “At this point, xxx cases per 100,000 persons were reported in south Africa when the survey were undertaken”.    
  5. Lines 181-184: it is not clear why “ordinal nature of the data” (format) can explain the data derivation (problem under investigation).
  6. Lines 206-208: this sentence is incomplete, between what and what?

Author Response

Dear Reviewer

Factors associated with limited vaccine literacy: lessons learnt from COVID-19 (vaccines-1680414)

Thank you for the valuable comments on our manuscript. Below is an indication of our response to each point raised. In addition, our corrections have been made in track changes in the manuscript.

1 - This is a questionnaire study that aimed to identify factors associated with limited vaccine literacy.  The survey collected data from 10,466 participants through an App Moya.  They reached the conclusion that 40% of respondents as having limited functional VL 292 compared to 8% of respondents with limited interactive-critical literacy.  Significant correlations of limited vaccine literacy were identified to be education, socio-economic status, vaccinated or not, race, health, and gender. This study is a well planned and well presented.  However, there are some shortcomings that need to be addressed.  One major concern is the unclear definition between the two criteria: functional and interactive-critical VL. For instance, on lines 271-276: people cannot understand the information appears to more relate to functional not interactive-critical (decision).  It is the author’s discretion to use any criteria in their study, but you need to make it clear the differences between the criteria you used.

Response:  We have rephrased lines 271-276 so that the definition of functional and interactive VL, as mentioned earlier in the manuscript comes through more clearly here.

Given the timing of our study, one might have expected even higher scores for functional VL as more information was available about COVID-19 vaccines, as well evidence that the vaccines were protecting individuals from infection. On the negative side, levels of functional VL may have remained lower as information about the vaccines are generally available in English, and respondents whose home language was not English might have struggled to read and understand the information. A higher score for interactive-critical VL suggests that our respondents were engaging with COVID-19 vaccine related-information and using it to make decisions, however as their functional VL was lower, they might not always have understood the information that they were engaging with (lines 287-295).

2 - Another major concern is the incomplete information to estimate the applicability of the results in this study.  Two pieces of information should be provided as clear as possible: (1) the percentage of the population who have access to the survey APP Moya. (2) has any incentives been used to stimulate the responses; this will give an indication which group are more likely to participate.

Response: At the time of the research, the Moya App had five million subscribers. However, we not only used the Moya App, but also Facebook, Twitter and word of mouth, to advertise the study. Therefore, we cannot determine how many peoples saw or heard about the survey. We have now included the following information in the limitations of the study:

Secondly, our sample was not randomly selected and as such we need to practice caution when interpreting and generalising our results (lines 390-392).

Respondents did not receive a direct incentive for participating in our study. We have now included the following information in the ethics section:

It was also indicated that while respondents would receive no remuneration for their participation, for every 1,000 completed questionnaires, 100 masks would be donated to a previously disadvantaged primary school, to a maximum of 1,000 masks (lines 153-156).

Minor concerns:

1 - The terminology “vaccine uptake” is ambiguous. In drug delivery, this means the efficiency of the drug delivery.  To avoid potential confusion, I suggest changing it to a more straightforward phase, like “vaccination rate” or “vaccination willingness” or sth others.

Response: The term “vaccine uptake” has been changed to “vaccination uptake” throughout the manuscript.

2 - Lines 54-56: this sentence does not read well. Have you missed “as” before “technology”?

Response: We have included ”as” before “technology”:

Misinformation about COVID-19 is prolific [13–14], for the first time in history, as technology and social media are being used on a massive scale to keep the public informed about this pandemic (lines 57-59).

3 - Lines 65-67: it is not clear from this sentence whether the quoted definition is VL or health literacy.

Response: The sentence has been changed as follows to make:

Vaccine literacy (VL) emanates from the concept of health literacy, which is defined as the “degree to which people have the capacity to obtain, process, and understand basic health information and services to make appropriate health decisions” [18] (p.228) (lines 68-70).

Thereafter vaccine literacy is now defined:

According to the same author, vaccine literacy does not just include knowledge about vaccines, but is also about “developing a system with decreased complexity to communicate and offer vaccines as sine qua non of a functioning health system” (lines 71-73). 

4 - Lines 86-87: this sentence is disconnected from the sentences above. It is suggested to change to “At this point, xxx cases per 100,000 persons were reported in south Africa when the survey were undertaken”.

Response: Note that the source document does not provide the number per 100,000 population rate. However, we have now added the approximate total number of people:

At this point, approximately 8.6 million people (more than one-fifth of the adult population) were fully vaccinated in the country [23] (lines 92-94).

5 - Lines 181-184: it is not clear why “ordinal nature of the data” (format) can explain the data derivation (problem under investigation).

Response: The ordinal nature of the data explains the “deviations” (not “derivations”)  from normality (line 192). Ordinal data is generally not normally distributed.

6 - Lines 206-208: this sentence is incomplete, between what and what?

Response: The sentence has been rephrased:

There was a significant difference between the mean scores for functional VL (M=2.841, SD 0.799) and interactive-critical VL (M=3.331, SD 0.559), t(10465)=51.403, p=0.000 (lines 216-218).

Please do not hesitate to contact with me with any other queries.

Yours sincerely

Michelle Englbrecht

Director & Senior Researcher: Centre for Health Systems Research & Development

Round 2

Reviewer 1 Report

Thanks

Author Response

Dear Reviewer

Thank you for your feedback. We have read and language edited the manuscript again (please see track changes in the attached document). 

Kind regards

Michelle Engelbrecht 

Reviewer 2 Report

I appreciate the authors's responses to the authors' concerns. I now understand the authors' definition of "coloured" as  an ethnicity, searching for it online it seems the term raises concerns that may vary between countries.

https://www.bbc.com/news/newsbeat-30999175

I understand it does not have the same meaning in South Africa. Because of its sensitivity abroad, I suggest the authors change the term to "Mixed Ethnicity". Otherwise I have no other concerns regarding the paper

Author Response

Dear Reviewer 

Thank you for your feedback. After careful consideration we would like to keep in the word "Coloured" as this terminology is used in official documents in our country as well as when describing socio-demographic characteristics of our population. We have also searched international journals and have found the word to be used in articles in journals such as Lancet Global Health as well as Social Science and Medicine. 

We have included the words "Coloured (i.e. people of mixed ethnic descent)" in the abstract. We have also elaborated on the explanation in the text "i.e., multiple ethnic groups with mixed ancestors from Africa, Asia and Europe". 

I hope that this meets with your approval.

Kind regards

Michelle Engelbrecht

Reviewer 3 Report

The authors has addressed all my concerns.

Author Response

Dear Reviewer

Thank you for your feedback. We have read the manuscript again and made minor language edits. 

Kind regards

Michelle Engelbrecht